# Cut Points of the Conicity Index and Associated Factors in Brazilian Rural Workers

**DOI:** 10.3390/nu14214487

**Published:** 2022-10-25

**Authors:** Camila Bruneli do Prado, Cleodice Alves Martins, Ana Clara Petersen Cremonini, Júlia Rabelo Santos Ferreira, Monica Cattafesta, Juliana Almeida-de-Souza, Eliana Zandonade, Olívia Maria de Paula Alves Bezerra, Luciane Bresciani Salaroli

**Affiliations:** 1Graduate Program Collective Health, Health Sciences, Federal University of Espírito Santo, Espírito Santo 29075-910, ES, Brazil; 2Graduate Program Nutrition and Health, Health Sciences, Federal University of Espírito Santo, Espírito Santo 29075-910, ES, Brazil; 3Nutrition’s Student, Federal University of Espírito Santo, Espírito Santo 29075-910, ES, Brazil; 4Centro de Investigação de Montanha (CIMO), Laboratório Associado para a Sustentabilidade e Tecnologia em Regiões de Montanha (SusTEC), Instituto Politécnico de Bragança, Campus de Santa Apolónia, 5300-25 Bragança, Portugal; 5Departament of Family Medicine, Mental and Collective Health, Medical School, Federal University of Ouro Preto, Ouro Preto 35400-000, MG, Brazil; 6Graduate Program Collective Health and Graduate Program Nutrition and Health, Health Sciences, Federal University of Espírito Santo, Espírito Santo 29075-910, ES, Brazil; lucianebresciani@gmail.com

**Keywords:** rural health, abdominal obesity, metabolic syndrome

## Abstract

(1) Background: Metabolic syndrome is associated with cardiovascular complications. Therefore, this study aims to establish cut points for the conicity index based on the components of metabolic syndrome and to associate it with characteristic sociodemographic, food consumption and occupational factors in Brazilian rural workers; (2) Methods: A cross-sectional study carried out with farmers. The receiver operating characteristic curve was calculated and the cut-off points for the conicity index were identified by the area under the curve, sensitivity and specificity. The variables included in the binary logistic regression analysis were selected by considering *p* < 0.20 in the bivariate test; (3) Results: The cut points were similar in females according to both criteria, resulting in a single cut-off of 1.269. In males, the cut points showed differences, resulting in 1.272 according to the NCEP-ATP III and 1.252 according to the IDF. We have shown that younger people, those who work more than 40 h a week and the lowest contribution of culinary ingredients are associated with increased odds of abdominal obesity, while the consumption of the products they sell or produce decreases these chances; (4) Conclusions: The conicity index showed high discriminatory power for the identification of abdominal obesity in rural workers. Therefore, there is a need to improve eating habits and promote healthier eating environments for individuals, respecting traditional food culture, mainly to contain the advance of MS in rural areas.

## 1. Introduction

Metabolic syndrome (MetS) is defined by a set of abnormalities, such as arterial hypertension, dyslipidemia, insulin resistance and obesity [1]. In Brazil, a multicenter epidemiological survey recorded a prevalence of MetS of 38.4%, with a high proportion in women, elderly individuals and less educated individuals [2]. However, studies on the health of rural workers are still scarce in the scientific literature, with those related to conditions intrinsic to work in the field being more predominant. This, added to the difficult access to primary health care and lower availability of services and specialized professionals, ends up generating underreporting of diagnosed diseases and mortality rates [3].

A recent study by Luz et al. [4] showed that six out of ten farmers in a region of southeastern Brazil had at least one cardiovascular risk factor, with arterial hypertension and dyslipidemia being the most prevalent. It is also worth noting that being overweight more than doubled the farmers’ chances of having two or more cardiovascular risk factors. In this same population, a 33.8% prevalence of insulin resistance was found, with obesity or being overweight increasing the risk of an individual presenting this condition by about three times [5].

The role of obesity as a potential metabolic risk factor is already well defined in the literature [6,7,8,9,10]. In a recent study with Brazilian farmers, weight was assessed from the body mass index (BMI) and was found to be higher than the general population [11]. It is known that BMI is an anthropometric indicator widely used to categorize nutritional status, however, it is limited by the fact that it does not provide the distribution of body fat [12,13,14]. Thus, the conicity index appears as an alternative measure to assess abdominal obesity [15].

The performance of the conicity index as an indicator of abdominal obesity from the MetS components has been demonstrated in numerous studies, but there is still no consensus on the cut points, which vary according to population, sex and age [1,2,3,4,5,6,7,8,9,10,11,12,13,14,15,16,17]. Until then, no cut point has been determined for this index in the Brazilian rural population, which justifies the importance of the present study.

Considering the risks to which rural workers are exposed, the difficulty in accessing health services and the high prevalence of multimorbidity in this population [18], it is necessary to establish a cut point for the index of conicity in order to generate an instrument capable of discriminating abdominal fat that can be used not only as a support material for policies and programs aimed at the health of rural workers, but also as a tool to be used by health professionals in the routines involved in primary care in rural areas.

Therefore, this study aims to establish cut points for the conicity index based on the components of metabolic syndrome according to the International Diabetes Federation (IDF) and the National Cholesterol Education Program’s Adult Treatment Panel III (NCEP-ATP III), and to associate it with characteristic sociodemographic, food consumption and occupational factors in Brazilian rural workers.

## 2. Materials and Methods

### 2.1. Study Design and Population

This is a cross-sectional study included in a larger project entitled “Health condition and associated factors: a study with farmers in Espírito Santo—AgroSaúdES”, funded by the Fundação de Amparo à Pesquisa do Espírito Santo (FAPES). The study was approved by the Research Ethics Committee of the Health Sciences Center of the Federal University of Espírito Santo, opinion n° 2091172 (CAAE 52839116.3.0000.5060) and meets the requirements required by Resolution of the National Health Council n. 466/12 and its complements for research involving human beings.

The study was carried out in the municipality of Santa Maria de Jetibá located in the state of Espírito Santo, southeastern Brazil, and it had a representative sample of farmers who met the following inclusion criteria: age between 18 and 59 years, not being pregnant, having agriculture as the main source of income and being in full employment for at least six months.

In the sample calculation, the population of farmers in the region (N = 7287), the expected prevalence of abdominal obesity of 50% and a significance level of 5% were considered. The minimum sample size was 365. However, as a way of improving the representativeness of the sample, data from 781 farmers who were investigated in the original project and who answered questions about the investigated outcome were analyzed.

Data collection took place between December 2016 and April 2017 on the premises of the municipality’s health units by properly trained researchers. The details involved in the data collection and research development are detailed in the article by Petarli et al. [18].

### 2.2. Data Collection

Farmers underwent blood collection for biochemical tests. As a way of minimizing errors, the analyzes were performed by a single laboratory and the fasting time required to perform the exams was 12 h.

The determination of HDL cholesterol was determined by the enzymatic colorimetric method with the Cholesterol Liquicolor Kit (In Vitro Diagnostica Ltd.a, Belo Horizonte, MG, Brazil) and the Cholesterol HDL Precipitation Kit (In Vitro Diagnostica Ltd.a, Belo Horizonte, MG, Brazil). Triglycerides were determined by the enzymatic colorimetric method with the Liquicolor mono^®^ Triglycerides Kit (In Vitro Diagnostica Ltd.a, Belo Horizonte, MG, Brazil).

Systolic blood pressure (SBP) and diastolic blood pressure (DBP) were measured during the interview 3 times for each subject using the Omron^®^ HME-7200 Automatic Pressure Monitor, calibrated and validated by the National Institute of Metrology, Quality and Technology (INMETRO). In order not to interfere with the results, the individuals were instructed to remain seated and rest for about 5 min, empty their bladders and not consume food, alcohol, coffee or cigarettes in the 30 min prior to the assessment. For data analysis, the average of two measurements was considered and a third measurement was performed whenever the difference between the first two was greater than 4 mmHg [18,19].

Food consumption data were obtained by applying three 24-h recalls (R24h) during the interview, two days from the week and one day from the weekend within 15 days after the first R24h in the return interviews. After the registration of the food and acquisition of the calories, no exclusion was performed due to extremes in energy consumption [20]. After obtaining the values of each R24h, the analysis of the attenuation was performed using the PC-SIDE software (Department of Statistics, Iowa State University, Ames, IA, USA), which follows the methodology of Nusser et al. [21]. Then, the adjustment for energy by the residual method was carried out, which corrects the estimates of nutrients by total energy intake [20]. The foods consumed by farmers in the three R24h were listed (n = 355 food items). These foods were classified according to NOVA, that is, in four groups: in natura and minimally processed foods; processed culinary ingredients; processed foods; and ultra-processed foods [22,23,24]. After the classification of food items in each NOVA group, the calories from each food group and subgroup were added. Then, the caloric contribution of each food group to daily energy consumption was calculated [23,25]. The energy contribution of each food group was categorized as “lower contribution” when below the median and “higher contribution” when above the median. More information about this collection and classification can be found in Cattafesta et al. [26].

Weight measurement was performed with participants barefoot, in an upright position, wearing as little clothing as possible [27] using a portable Tanita^®^ scale. Height was measured with the individuals barefoot, standing, in an upright position, arms extended along the body and eyes fixed on a point on the horizon [27] using a portable Sanny^®^ stadiometer. Waist circumference (WC) was measured with the participant standing, with arms extended along the body and feet together, with the inelastic tape positioned at the midpoint between the lower edge of the costal arch and the iliac crest [27]. Three non-consecutive measurements were taken, the first being discarded and the average of the last two considered as the final measurement. In relation to BMI, it was categorized considering the WHO cut points [28] in eutrophy/low weight and overweight/obesity.

The conicity index was calculated from the measurements of body mass, height and WC through the following mathematical equation [15]:$$\mathrm{Conicity}\;\mathrm{Index}=\frac{\mathrm{Waist}\;\mathrm{circumference}\;(m)}{0.109\sqrt{\frac{\mathrm{Body}\;\mathrm{mass}\;(\mathrm{kg})}{\mathrm{Stature}\;(m)}}}$$

To establish the index cut-off point for both sexes, the MS criteria according to the IDF and NCEP-ATP III were used. In the first, the individual with abdominal obesity is considered to have the syndrome, assessed by a WC ≥ 84 cm for women or ≥94 cm for men, occurring simultaneously with two more criteria: fasting blood glucose ≥ 100; SBP ≥ 130 mmHg or DBP ≥ 85 mmHg; TG ≥ 150 mg/dL and HDL-c < 40 mg/dL for men and <50 mg/dL for women. The use of antihypertensive, hypoglycemic and/or lipid-lowering drugs are considered in both criteria for MS, as they classify the individual with hypertension, diabetes and/or dyslipidemia, respectively [29]. The NCEP considers MS in the presence of at least three of the following criteria: WC > 102 cm for men or >88 cm for women; HDL-c < 40 mg/dL for men and <50 mg/dL for women; TG ≥ 150 mg/dL; SBP ≥ 130 mmHg and DBP ≥ 85 mmHg and fasting blood glucose ≥ 100.

### 2.3. Statistical Analysis

To describe the study variables, absolute and relative frequencies were used. To verify if there was a difference between the proportions of the independent variables and the outcome, Pearson’s chi-square test (x²) was used for qualitative variables.

The data were submitted to analysis of the ROC curve (Receiver Operating Characteristic) to establish the cut points for the conicity index according to the set of conditions that make up the criteria for MetS in both diagnostic criteria mentioned above. Cut points were defined based on accuracy, specificity and sensitivity.

A hierarchical logistic regression was performed for the association of the conicity index with the independent variables, including the variables that presented a value of *p* ≤ 0.20 in the bivariate analysis. From the first to the fourth models, socioeconomic, behavioral, food consumption, anthropometric and work variables were aggregated, respectively. Among the sociodemographic variables evaluated were sex, age group (“up to 29 years”, “30 to 39 years”, “40 to 49 years” and “50 years or more”), schooling (“less than 4 years”, “4 to 8 years” and “more than 8 years”), marital status (“unmarried”, “married or lives with a partner” and “divorced or widowed”), race/color (“white” and “non-white”), land bond (“owner” and “non-owner”) and socioeconomic class (“A or B”, “C” and “D or E”), according to the Criteria of Economic Classification Brazil. Labor variables were investigated by questioning the current type of production (“conventional” and “non-conventional”) and the workload (hours/week) (“less than or equal to 40 h” and “more than 40 h”). Lifestyle variables included alcohol consumption, categorized as “non-drinking” and “drinking”; smoking, assessed according to the Smoker Approach and Treatment Consensus and categorized as “non-smoker” and “current and past smoker”; practice of physical activity extra-field (“yes” or “no”). As a behavioral variable, we used the consumption of the food produced or sold, categorized as “yes” or “no”. Among models 1 to 3, the enter method was used. The final model was performed using the Forward LR method. For all of them, the assumptions of absence of multicollinearity and absence of outliers were respected.

Odds ratios (OR), adjusted 95% confidence intervals (CI) and a 5% significance level were presented. All analyzes were conducted in R software (4.0.3) for Windows. The significance level adopted was 5%.

## 3. Results

The result of the analysis of the ROC curve of the conicity index, according to sex, based on the NCEP criterion can be seen in Figure 1. It is noted that for women, the conicity index has an area under the curve (AUC) corresponding to 0.804 (95% CI: 0.748–0.860, *p* < 0.001), while in men the AUC was 0.850 (95% CI: 0.787–0.913, *p* < 0.001) (Figure 1).

Figure 2 shows the result of the ROC curve based on the IDF criterion in both sexes. It is observed that in women, the conicity index has an AUC of 0.784 (95% CI: 0.728–0.840, *p* < 0.001), while in men it was 0.845 (95% CI: 0.796–0.895, *p* < 0.001).

The cut points were similar in females according to both criteria, resulting in a single cut-off of 1.269. Differences were found in the accuracy criteria: NCEP-ATP III with sensitivity and specificity of 83.60% and 69.30%, respectively, and IDF with sensitivity and specificity of 77.3% and 70.2%, respectively. In males, the cut points showed differences, resulting in 1.272 according to the NCEP-ATP III (sensitivity and specificity of 83.30% and 74.90%, respectively) and 1.252 according to the IDF (sensitivity and specificity of 88, 90% and 69.70%, respectively) (Table 1).

Table 2 presents the bivariate analysis of indicators in relation to sociodemographic, lifestyle and occupational variables. For the IDF criterion, there are proportional differences between age (*p* < 0.001), education (*p* < 0.001), marital status (*p* = 0.010), smoking (*p* = 0.032), physical activity (*p* = 0.008) and BMI (*p* < 0.001). In the NCEP criterion, the variables that showed differences were age (*p* < 0.001), education (*p* < 0.001), marital status (*p* = 0.022), alcohol consumption (*p* = 0.030), physical activity (*p* = 0.013) and BMI (*p* < 0.001).

The hierarchical logistic regression between the MetS components according to IDF and the conicity index can be seen in Table 3. It was found that age ≤29 years (OR: 11.31; *p* < 0.001; 95% CI: 5.82–21.98), 30 to 39 years (OR: 4.53; *p* < 0.001; 95% CI: 2.80–7.34) and 40 to 49 years (OR: 1.82; *p* = 0.009; 95% CI: 1.15–2.88) and working more than 40 h a week (OR: 1.74; *p* = 0.009; 95% CI: 1.14–2.67) were associated with the presence of abdominal fat. Regarding the degree of processing of culinary ingredients, the reduction in consumption of these ingredients was 1.66 times more likely to have a metabolic risk (OR: 1.66; *p* = 0.004; 95% CI: 1.17–2.35).

Table 4 shows the results of the hierarchical logistic regression between the MetS components according to NCEP-ATP III and the conicity index. Age ≤29 years (OR: 12.51; *p* < 0.001; 95% CI: 6.46–24.25), 30 to 39 years (OR: 4.75; *p* < 0.001; 95% CI: 2.94–7.68) and 40 to 49 years (OR: 1.86; *p* = 0.006; 95% CI: 1.19–2.92) and reduction in consumption of culinary ingredients (OR: 1.57; *p* = 0.008; 95% CI: 1.12–2.22) were associated with increased abdominal fat, while consuming the products they sell or produce reduced the chances of having abdominal fat by 64% (OR: 0.36; *p* = 0.040; 95% CI: 0.13–0.95).

## 4. Discussion

This study is a pioneer in identifying the cut-off point and showing the effectiveness and reliability of the obesity index as evidence of MetS stratified by sex in rural workers. The cut-off points found can be used in epidemiological studies and in clinical practice in this specific population.

The conicity index was proposed by Valdez [15] as an anthropometric indicator to identify the distribution of body fat, specifically abdominal adiposity. However, it presents difficulties in its routine use due to the absence of standardized cut-off points for reference in the population. According to the mathematical equation that gives rise to the index, the denominator determines the shape of the body by considering the height and weight of individuals. This is an advantage over other anthropometric indicators, as it allows comparison between different groups [15].

The cut point of the conicity index in Brazilians most used in the literature was performed in healthy adults to discriminate the high coronary risk, presenting values of 1.25 and 1.18 for men and women, respectively [30]. In addition to this study, other authors identified high values when they used the diagnostic criteria for MetS, between 1.61 and 1.56 for men and women, respectively [31], and 1.36 for women [32]. When compared with the values found in this study, a divergence is observed mainly in females, highlighting the need to use the specific cut point for rural workers.

Regarding age, both diagnostic criteria (IDF and NCEP-ATP III) showed results that indicate a higher risk for the presence of abdominal fat in younger individuals. This finding is already consolidated in the literature, showing that younger individuals have greater general and abdominal obesity compared to the elderly [33], and in rural areas it was shown that the younger population tended to consume more ultra-processed foods [26].

The NOVA classification system was first presented by Monteiro et al. [34] and classifies all foods into four groups according to the nature, extent and purpose of industrial food processing, namely: Group 1—foods unprocessed/minimally processed; Group 2—processed culinary ingredients; Group 3—processed foods; Group 4—ultra-processed foods [34]. Regarding culinary ingredients, these are used to season and cook food and create culinary preparations [22]. The reduction in the consumption of foods with a low degree of processing is a parameter that confers a risk of abdominal obesity above normal.

With regard to our findings, as per the IDF and NCEP classification, reducing the consumption of culinary ingredients increases the risk of high abdominal fat by 1.66-fold and 1.57-fold, respectively. This finding is in line with the literature since other studies, including a study carried out with this same population, found that lower adherence to traditional dietary patterns is associated with a higher prevalence of both general obesity and abdominal obesity [11,35,36,37]. Historically, the food pattern of rural populations consists of the consumption of rice, beans, cassava flour, coffee, cow’s milk and bread, but in recent years there has been an increase in the consumption of traditional industrialized products in urban areas, which highlights a new scenario with the influence of globalization [26,38,39,40].

In the young adult population, Santana et al. [41] relate the consumption of ultra-processed foods to the presence of abdominal obesity due to their poor nutritional quality and higher total energy value, which is one of the cardiometabolic risk factors in the progressive increase in these individuals. Concomitantly with obesity, there has also been an increase in the development of hypertension, diabetes and cardiovascular diseases, chronic non-communicable diseases typical of a diet rich in industrialized foods that are the result of metabolic abnormalities, often caused by inflammation resulting from excess sugars and fats in ultra-processed foods [4,35,42,43].

In addition, this study showed that the consumption of food for sale is considered a factor of abdominal obesity. It is already established in obesity that fresh foods are less energetically dense and are rich in vitamins, minerals and fibers that act as protectors against the literature and cardiometabolic diseases [44,45,46]. It is noteworthy that among the main agricultural productions in this region are the planting of potatoes, beans, cassava, corn, tomatoes, bananas, coffee, guava, lemons, grapes, passion fruit and palm hearts [47] and many of these represent healthy choices important for this population. In a study carried out with the same sample, it was found that more than half of the population (64.9%) consumed tomatoes and that bananas and lemons were the fruits most consumed by the participants [6].

The limitation of the present study is its cross-sectional nature, which requires greater caution in interpreting the results due to the possibility of reverse causality. Despite this, the importance of the findings presented is justified, since the study showed high rigor in the sampling process, reaching a representative sample of the population of farmers. The lack of articles that use the same methodology in rural populations represents both a limitation and a strong point of the work.

It represents a limitation as it makes it difficult to compare the results, but it represents a strong point, because, in addition to its unprecedented nature, it may represent the beginning of new research aimed at the health of this population, especially the assessment of body adiposity, in order to reach a consensus on the future cut-off point and publicize the use of this technique in clinical practice.

## 5. Conclusions

In conclusion, cut-off points of 1.269 for women and 1.252 to 1.272 for men for the conicity index showed good sensitivity and specificity for identifying abdominal obesity in farmers. Thus, the conicity index proves to be an accurate and easily accessible marker to be incorporated into clinical practice in this population.

There is a need to improve eating habits and promote healthier eating environments for individuals, while respecting the traditional food culture, especially to contain the advancement of MetS in rural areas. Finally, due to the multicausal nature of MetS, coping strategies for this condition must include a multiple, intersectoral and interdisciplinary approach.

## Figures and Tables

**Figure 1 nutrients-14-04487-f001:**
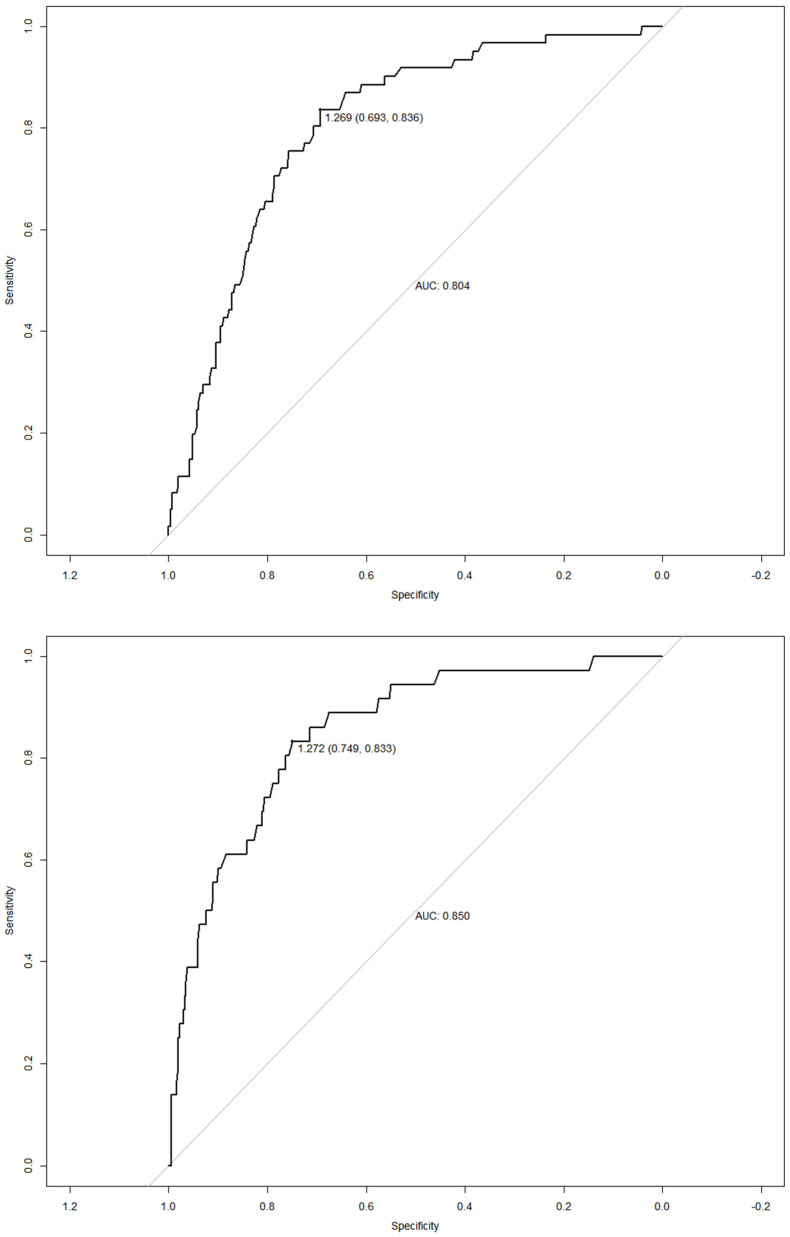
ROC curves of the conicity index for the diagnosis of abdominal obesity in female and male Brazilian rural workers according to the NCEP −ATP III criteria, respectively. AUC—area under the curve.

**Figure 2 nutrients-14-04487-f002:**
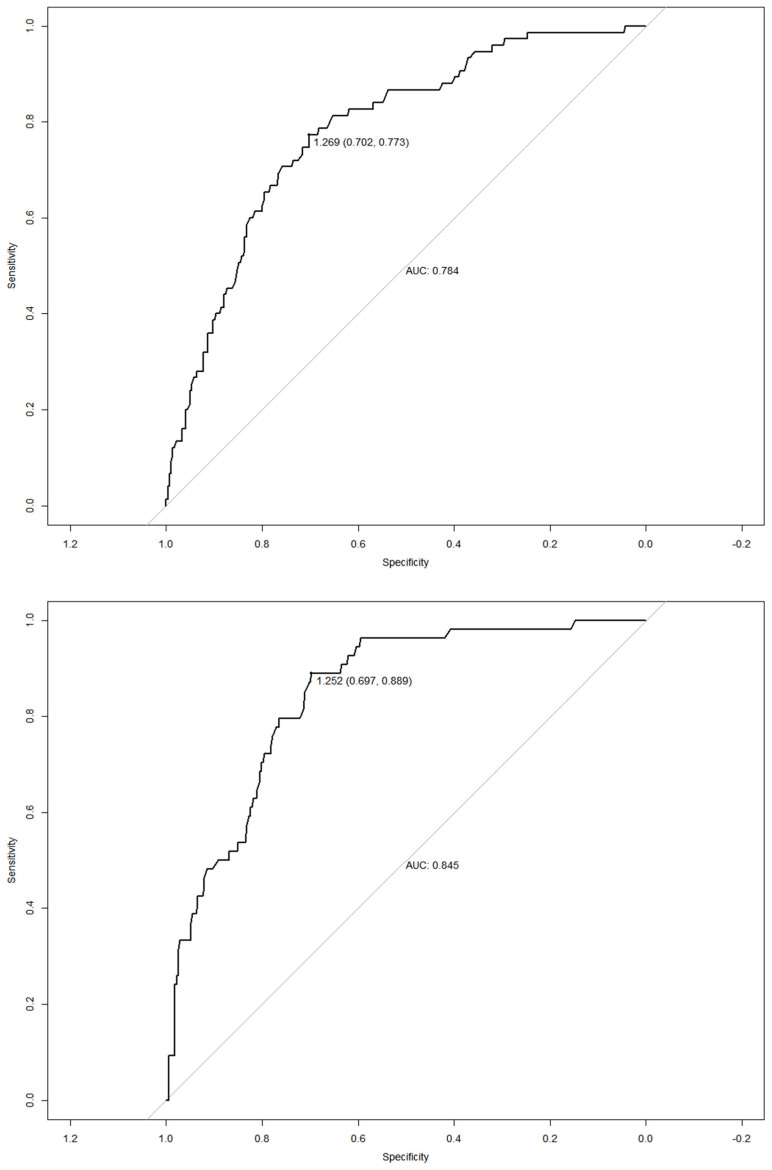
ROC curves of the conicity index for the diagnosis of abdominal obesity in female and male Brazilian rural workers according to the IDF criteria, respectively. AUC—area under the curve.

**Table 1 nutrients-14-04487-t001:** Cut points and diagnostic performance measures for the conicity index according to the MetS criteria.

Variables	NCEP—ATP III		IDF	
	Men (95% CI)	Women (95% CI)	Men (95% CI)	Women (95% CI)
Cut points	1.272	1.269	1.252	1.269
AUC ^1^	0.850 (0.787–0.913)	0.804 (0.748–0.860)	0.845 (0.796–0.895)	0.784 (0.728–0.840)
Accuracy	0.757 (0.756–0.758)	0.717 (0.716–0.718)	0.722 (0.721–0.73)	0.717 (0.716–0.718)
Sensitivity	0.833 (0.712–0.955)	0.836 (0.743–0.929)	0.889 (0.805–0.973)	0.773 (0.679–0.868)
Specificity	0.749(0.705–0.793)	0.693 (0.642–0.744)	0.697 (0.649–0.745)	0.702 (0.651–0.754)
PPV ^2^	0.244(0.168–0.320)	0.347 (0.270–0.424)	0.310 (0.237–0.382)	0.395 (0.316–0.474)
NPV ^3^	0.979(0.962–0.996)	0.956 (0.929–0.983)	0.976 (0.957–0.995)	0.925 (0.891–0.959)

^1^ AUC—area under the curve; ^2^ PPV—positive predictive value; ^3^ NPV—negative predictive value. 95% CI—95% confidence interval.

**Table 2 nutrients-14-04487-t002:** Sociodemographic, food consumption and occupational factors of rural works.

Variables	IDF			NCEP-ATP III		
	CI ^1^ Adequate	CI Elevated	*p*-Value	CI Adequate	CI Elevated	*p*-Value
Sex			0.836			0.014
Women	227 (29.06%)	147 (18.82%)		227 (29.06%)	147 (18.83%)	
Men	251 (32.13%)	156 (19.97%)		282 (36.10%)	125 (16.01%)	
Age (group)			<0.001			<0.001
≤29 years	153 (19.60%)	23 (2.94%)		155 (19.85%)	21 (2.69%)	
≥30 to 39 years	173 (22.15%)	70 (8.96%)		183 (23.45%)	60 (7.68%)	
≥40 to 49 years	98 (12.55%)	103 (13.19%)		108 (13.82%)	93 (11.90%)	
≥50 years or more	54 (6.91%)	107 (13.70%)		63 (8.06%)	98 (12.55%)	
Schooling (years)			<0.001			<0.001
<4 years	286 (36.62%)	240 (30.73%)		308 (39.44%)	218 (27.91%)	
4 to 8 years	129 (16.52%)	42 (5.37%)		132 (16.90%)	39 (5.00%)	
>8 years	63 (8.06%)	21 (2.70%)		69 (8.83%)	15 (1.92%)	
Marital status			0.010			0.022
Married or lives with partner	401 (51.34%)	270 (34.57%)		430 (55.05%)	241 (30.86%)	
Divorced or widowed	30 (3.84%)	21 (2.68%)		31 (3.96%)	20 (2.57%)	
Unmarried	47 (6.02%)	12 (1.53%)		48 (6.15%)	11 (1.41%)	
Self-referred race/color			0.281			0.324
No white	59 (7.55%)	29 (3.71%)		62 (7.94%)	26 (3.33%)	
White	419 (53.65%)	274 (35.08%)		447 (57.23%)	246 (31.50%)	
Socioeconomic class			0.774			0.973
Class A or B	35 (4.48%)	23 (2.94%)		37 (4.74%)	21 (2.68%)	
Class C	235 (30.09%)	156 (19.97%)		255 (32.65%)	136 (17.41%)	
Class D or E	208 (26.63%)	124 (15.87%)		217 (27.78%)	115 (14.72%)	
Land bond			0.179			0.324
Owner	359 (45.96%)	241 (30.85%)		385 (49.30%)	215 (27.53%)	
No owner	119 (15.23%)	62 (7.94%)		124 (15.87%)	57 (7.30%)	
Type of production			0.112			0.268
Conventional	438 (56.15%)	267 (34.23%)		464 (59.49%)	241(30.90%)	
No conventional	39 (5.00%)	36 (4.61%)		44 (5.64%)	31 (3.97%)	
Workload			0.057			0.003
≤40 h/week	87 (11.13%)	73 (9.34%)		88 (11.26%)	72 (9.22%)	
>40 h/week	391 (50.06%)	230 (29.44%)		421 (53.90%)	200 (25.60%)	
Alcohol intake			0.239			0.030
No	259 (33.16%)	178 (22.80%)		270 (34.57%)	167 (21.38%)	
Yes	219 (28.05%)	125 (16.00%)		239 (30.60%)	105 (13.44%)	
Smoking			0.032			0.365
No smoking	415 (53.13%)	245 (31.37%)		435 (55.70%)	225 (28.80%)	
Current smoking or past	63 (8.06%)	58 (7.42%)		74 (9.47%)	47 (6.02%)	
Physical activity off field			0.008			0.013
Do not practice	373 (47.76%)	263 (33.67%)		400 (51.21%)	236 (30.21%)	
Below recommended	63 (8.06%)	23 (2.94%)		67 (8.58%)	19 (2.43%)	
Within the recommended	42 (5.37%)	17 (2.17%)		42 (5.37%)	17 (2.17%)	
Body mass index			<0.001			<0.001
Low weight/eutrophy	339 (43.40%)	44 (5.63%)		349 (44.68%)	34 (4.35%)	
Overweight/obesity	139 (17.80%)	259 (33.16%)		160 (20.48%)	238 (30.47%)	
Consumption of minimally processed			0.476			0.795
Lower contribution	217 (29.68%)	148 (20.24%)		239 (32.01%)	127 (17.92%)	
Higher contribution	228 (31.20%)	138 (18.87%)		234 (32.69%)	131 (17.37%)	
Consumption of culinary ingredients			0.118			0.056
Lower contribution	233 (31.87%)	132 (18.05%)		249 (34.06%)	116 (15.86%)	
Higher contribution	212 (29.00%)	154 (21.06%)		224 (30.64%)	142 (19.42%)	
Consumption of processed			0.198			0.253
Lower contribution	212 (29.00%)	151 (20.65%)		227 (31.05%)	136 (18.60%)	
Higher contribution	233 (31.87%)	135 (21.20%)		246 (33.65%)	112 (16.68%)	
Consumption of ultra-processed			0.076			0.301
Lower contribution	235 (32.14%)	131 (17.92%)		244 (33.38%)	122 (16.69%)	
Higher contribution	210 (28.72%)	155 (21.20%)		229 (31.32%)	136 (18.60%)	
Consumption products you sell			0.341			0.098
No	22 (2.81%)	9 (1.15%)		25 (3.20%)	6 (0.76%)	
Yes	456 (58.38%)	294 (37.64%)		484 (61.97%)	266 (34.05%)	

^1^ CI—conicity index.

**Table 3 nutrients-14-04487-t003:** Hierarchical logistic regression between the associated variables in the bivariate analysis and the conicity index according to the components of the IDF.

Variables	Model 1		Model 2		Model 3		Final Model	
	*p*-Value	OR (CI _95%_)	*p*-Value	OR (CI _95%_)	*p*-Value	OR (CI _95%_)	*p*-Value	OR (CI _95%_)
Age (group)								
≤29 years	<0.001	10.06 (5.51–18.39)	<0.001	9.54 (5.20–17.51)	<0.001	11.31 (5.82–21.98)	<0.001	11.31 (5.82–21.98)
≥30 to 39 years	<0.001	4.61 (2.95–7.20)	<0.001	4.33 (2.76–6.80)	0.008	4.53 (2.80–7.34)	<0.001	4.53 (2.80–7.34)
≥40 to 49 years	0.005	1.83 (1.19–2.79)	0.010	1.74 (1.13–2.67)	0.009	1.82 (1.15–2.88)	0.009	1.82 (1.15–2.88)
≥50 years or more		1		1		1		1
Marital status								
Married or lives with partner		1		1		1		1
Unmarried	0.523	1.27 (0.60–2.64)	0.431	1.34 (0.64–2.82)	0.618	1.21 (0.56–2.57)	0.618	1.21 (0.56–2.57)
Divorced or widowed	0.256	1.43 (0.77–2.66)	0.186	1.52 (0.81–2.86)	0.201	1.51 (0.80–2.85)	0.201	1.51 (0.80–2.85)
Schooling (years)								
<4 years		1		1		1		1
4 to 8 years	0.540	1.15 (0.73–1.80)	0.465	1.18 (0.75–1.86)	0.608	1.13 (0.70–1.82)	0.608	1.13 (0.70–1.82)
>8 years	0.066	1.81 (0.96–3.41)	0.054	1.87 (0.98–3.56)	0.053	1.92 (0.99–3.73)	0.053	1.92 (0.99–3.73)
Bond with the earth								
No owner		1		1		1		1
Owner	0.636	0.91 (0.61–1.34)	0.413	0.84 (0.56–1.26)	0.448	0.84 (0.55–1.29)	0.448	0.84 (0.55–1.29)
Workload								
≤40 h/week				1		1		1
>40 h/week			0.007	1.72 (1.15–2.56)	0.009	1.74 (1.14–2.67)	0.009	1.74 (1.14–2.67)
Type of production								
Conventional				1		1		1
No conventional			0.537	0.84 (0.49–1.44)	0.374	0.77 (0.44–1.35)	0.374	0.77 (0.44–1.35)
Physical activity off field								
Do not practice						1		1
Below recommended					0.282	1.38 (0.76–2.52)	0.282	1.38 (0.76–2.52)
Within the recommended					0.830	1.07 (0.54–2.14)	0.830	1.07 (0.54–2.14)
Smoking								
No smoking						1		1
Current smoking or past					0.382	1.23 (0.76–1.99)	0.382	1.23 (0.76–1.99)
Degree of processing								
Culinary ingredients								
Higher contribution						1		1
Lower contribution					0.004	1.66 (1.17–2.35)	0.004	1.66 (1.17–2.35)
Processed								
Higher contribution						1		1
Lower contribution					0.759	0.94 (0.65–1.35)	0.759	0.94 (0.65–1.35)
Ultra-processed								
Higher contribution						1		1
Lower contribution					0.245	1.24 (0.86–1.79)	0.245	1.24 (0.86–1.79)

**Table 4 nutrients-14-04487-t004:** Hierarchical logistic regression between the associated variables in the bivariate analysis and the conicity index according to the components of the NCEP-ATP III.

Variables	Model 1		Model 2		Model 3		Final Model	
	*p*-Value	OR (CI _95%_)	*p*-Value	OR (CI _95%_)	*p*-Value	OR (CI _95%_)	*p*-Value	OR (CI _95%_)
Sex								
Women		1		1		1		1
Men	0.001	1.68 (1.21–2.33)	0.010	1.55 (1.10–2.18)	0.074	1.41 (0.96–2.07)	0.074	1.41 (0.96–2.07)
Age (group)								
≤29 years	<0.001	11.39 (6.18–20.98)	<0.001	10.98 (5.95–20.27)	<0.001	12.51 (6.46–24.25)	<0.001	12.51 (6.46–24.25)
≥30 to 39 years	<0.001	4.93 (3.14–7.76)	<0.001	4.73 (3.00–7.46)	<0.001	4.75 (2.94–7.68)	<0.001	4.75 (2.94–7.68)
≥40 to 49 years	0.003	1.90 (1.24–2.92)	0.004	1.85 (1.21–2.85)	0.006	1.86 (1.19–2.92)	0.006	1.86 (1.19–2.92)
≥50 years or more		1		1		1		1
Marital status								
Married or lives with partner		1		1			1	
Unmarried	0.795	1.10 (0.52–2.34)	0.681	1.17 (0.55–2.49)	0.865	1.06 (0.49–2.31)	0.865	1.06 (0.49–2.31)
Divorced or widowed	0.147	1.58 (0.84–2.96)	0.123	1.64 (0.87–3.07)	0.157	1.58 (0.83–2.99)	0.157	1.58 (0.83–2.99)
Schooling (years)								
<4 years		1		1		1		1
4 to 8 years	0.645	1.11 (0.70–1.74)	0.607	1.12 (0.71–1.77)	0.679	1.10 (0.68–1.78)	0.679	1.10 (0.68–1.78)
>8 years	0.083	1.74 (0.92–3.29)	0.079	1.76 (0.93–3.32)	0.126	1.66 (0.86–3.19)	0.126	1.66 (0.86–3.19)
Workload								
≤40 h/week				1		1		1
>40 h/week			0.088	1.42 (0.94–2.13)	0.055	1.53 (0.98–2.36)	0.055	1.53 (1.01–2.36)
Physical activity off field								
Do not practice						1		1
Below recommended					0.302	1.37 (0.75–2.50)	0.302	1.37 (0.75–2.50)
Within the recommended					0.848	1.06 (0.53–2.13)	0.848	1.06 (0.53–2.13)
Alcohol intake								
No						1		1
Yes					0.679	1.08 (0.74–1.56)	0.679	1.08 (0.74–1.56)
Degree of processing								
Culinary ingredients								
Higher contribution						1		1
Lower contribution					0.008	1.57 (1.12–2.22)	0.008	1.57 (1.12–2.22)
Consumption products you sell								
No						1		1
Yes					0.040	0.36 (0.13–0.95)	0.040	0.36 (0.13–0.95)

OR—odds ratio; 95% CI—confidence interval 95%.

## Data Availability

Not applicable.

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
