# Peer review of "Cut Points of the Conicity Index and Associated Factors in Brazilian Rural Workers"

_nutrients, 2022, doi:10.3390/nu14214487_

Round 1
Reviewer 1 Report
In the manuscript „Cut points of the conicity index and associated factors in Brazilian rural workers: the Authors tried to establish cut points for the conicity index based on the components of the metabolic syndrome, and to associate it with characteristics of sociodemographic, food consumption and occupational factors in Brazilian rural workers. This is a cross-sectional study, which included 781 farmers aged 18-59 years from Brazil. The Authors tried to fill the gap in regard to identify the cut-off point and showing the reliability and reliability of the obesity index stratified by sex in rural workers. The results of the study can be used in epidemiological studies and in clinical practice in this specific population.
Generally, the manuscript provides valuable information. In the introduction, there is a brief description of the problem and the purpose of the study. The Authors very clearly described materials and methods. Moreover, the Authors have got approval from the Research Ethics Committee of the Health Sciences Center of the Federal University of Espírito Santo. The Authors used proper procedure in data analysis. The results are described well and concisely. The discussion is conducted in an interesting way, and the conclusions relate to the study.
Reviewer 2 Report
The paper highlights high discrimination of abdominal obesity amongst rural workers which is a risk factor for cardiovascular diseases . The study provides this insight using the conicity index. I would like to recommend minor corrections.
- Were hours of working included in the variables?
- “Reliability” word repeated in line 241 and 242.
- The discussion could highlight the specific cardiovascular disorders in younger individuals and its association with ultraprocessed foods.
- Wrong parenthesis “)” in line no 277.
